Electric resistance tomography and stress wave tomography for decay detection in trees—a comparison study

Yue Xiaoquan 1
Wang Lihai wanglihai@nefu.edu.cn wanglihai2012@126.com 2
Wacker James P. 3
Zhu Zhiming 1
1 College of Transportation and Civil Engineering, Fujian Agriculture and Forestry University of China , Fuzhou , China
2 College of Engineering and Technology, Northeast Forest University , Harbin , China
3 Forest Service, Forest Products Laboratory, United States Department of Agriculture , Madison, WI , United States of America
Marino Bruno
Electronic publication date: 2019 Mar 5
Publication date: 2019
Volume: 7
Electronic Location ID: e6444
Received 2018 Oct 9; Accepted 2019 Jan 14
Copyright: ©2019 Yue et al.
Copyright year: 2019
Copyright holder: Yue et al.
License: This is an open access article distributed under the terms of the Creative Commons Attribution License, which permits unrestricted use, distribution, reproduction and adaptation in any medium and for any purpose provided that it is properly attributed. For attribution, the original author(s), title, publication source (PeerJ) and either DOI or URL of the article must be cited.
License URL: https://creativecommons.org/licenses/by/4.0/

Keywords: Stress wave tomography, Mass loss, Electric resistance tomography, The severity of decay, Nondestructive techniques in live trees

Funding: National Natural Science Foundation of China No. 31570547 National “948” project of China No. 2014-4-78 Science and Technology Innovation Special Fund Project of Fujian Agriculture & Forestry University No. KFA17039A This work was supported by the National Natural Science Foundation of China (No. 31570547), the fund of national “948” project of China (No. 2014-4-78), and the Science and Technology Innovation Special Fund Project of Fujian Agriculture & Forestry University (No.KFA17039A). The funders had no role in study design, data collection and analysis, decision to publish, or preparation of the manuscript.

==============================
Background

To ensure the safety of trees, two NDT (nondestructive testing) techniques, electric resistance tomography and stress wave tomography, were employed to quantitatively detect and characterize the internal decay of standing trees. Comparisons between those two techniques were done to make full use of the individual capability for decay detection.

Methods

Eighty trees (40 Manchurian ash and 40 Populus simonii) were detected, then wood increment cores were obtained from each cross disc trial. The Dt, which was defined as the value determined by the mass loss ratio of each wood core, was regarded as the true severity of decay. Using ordinary least-squares regression to analyze the relationship between Dt and De (De was defined as the severity of decay determined by electric resistance tomography) and between Dt and Ds (Ds was defined as the severity of decay determined by stress wave tomography).

Results

The results showed that both methods could estimate the severity of decay in trees. In terms of different stages of decay, when Dt < 30%, De had a strong positive correlation with Dt (R2 = 0.677, P < 0.01), while, when Dt ≥ 30%, Ds had a significant positive correlation relationship with Dt (R2 = 0.645, P < 0.01).

Conclusion

Electric resistance tomography was better than stress wave tomography for testing in the early stages of decay, while stress wave tomography can be used effectively in the late stage of decay. It is suggested that each technique can be used in the practice of internal decay testing of standing trees based on decay stages and operating conditions.

Introduction

Nondestructive testing (NDT) can detect the decay of wood quickly and accurately without damaging the wood (Pellerin & Ross, 2002; Fang, Lujun & Feng, 2017). Along with the characteristic index and diagnosis of the internal condition of wood, NDT can provide a scientific basis for assessment of standing trees and can guide forest management, as well as provide an important reference for bucking and processing of wood. In recent years, a variety of non-destructive evaluation techniques have been used to investigate and detect the internal decay of standing trees (Brashaw et al., 2009; Ruz, Estevez & Ramirez, 2009; Proto et al., 2017). Practical applications show that each technology has its advantages and disadvantages. Therefore, it is necessary to compare these technologies to determine appropriate technologies that are suitable for the specific survey conditions of the trees in the forest.

Electric resistance tomography (ERT) is a rapidly developed technique for wood defect detection in recent years (Al Hagrey, 2006; Al Hagrey, 2007; Humplík, Čermák & Žid, 2016). The principle of ERT technique is that when the instrument opens the test switch, current excitation is generated at the test cross-section, and the peripheral voltage of the trunk can be measured. The discrete network in the cross section is calculated by the specific algorithm inside the instrument. The resistance value of each point after the gridding and the different values are assigned to different pixels after digital image processing is output, that is, a 2D (two-dimensional) image of resistance detection is established (Bertallot, Canavero & Comino, 2000). The application of ERT in wood defect detection has been proved (Just & Jacbbs, 1998; Yue et al., 2018). In combination with acoustic computed tomography and electric impedance tomography ( Brazee et al., 2011), ERT has been used to detect and quantify the internal decay of standing trees. As a measure of electrical resistance, ERT can be used to analyze moisture distribution and movement in the tree trunk (Xu, Xu & Wang, 2014; Nadler & Tyree, 2008). ERT also has been used to evaluate tree trunk decay or the sapwood-heartwood interface in dicotyledonous and coniferous trees (Guyot et al., 2013; Nicolotti et al., 2003; Lin et al., 2012).

Stress wave technology is commonly used as a nondestructive testing technique for wood (Robert et al., 2005; Yang et al., 2017). Stress wave tomography (SWT) is a two-dimensional image formed by the relative velocity of stress wave propagation to reflect the internal conditions of the wood. The specific process is to use a pulse hammer to knock the stress wave sensor fixed on the tree trunk to make the propagation of stress mechanical waves inside the tree trunk, by measuring the time at which other sensors receive the wave signal, and converting it into the corresponding direction of the propagation velocity, and then reconstructing by the wave velocity matrix transformation to form a two-dimensional image of the measured cross section, so it can intuitively reflect the internal conditions of the wood section (Huang et al., 2013). Due to the ability to detect mechanical properties and internal defects, stress wave tomography is widely used to detect wood defects. The application and propagation rules of SWT were studied by detecting defects in trees and assessing the safety conditions of trees (Wang et al., 2007). The number of sensors can influence the fit and error rate of stress wave images, at least 12 sensors are needed to make the image fit close to 90%, and the error rate is reduced to 0.1 (Wang, Xu & Zhou, 2007). Study has indicated that the relationship between stress waves and resistance values was significant (Allison & Wang, 2015).

However, to date, the exact technology that matched the specific investigation conditions of standing trees in the forest has not yet been accurately found. In this paper, therefore, we aim to make comparisons between ERT and SWT. On the basis of field and laboratory tests, the decay of standing trees trunks was detected, and the different stages of decay were quantitatively described. Three methods were used to test the cross-section of the standing wood: ERT, SWT, and mass loss of wood increment cores. Taking wood core samples as research objects, the results of the other two non-destructive testing methods were compared to determine the appropriate technology suitable for conditions of standing trees in the forest.

Materials and Methods

Field testing

The study was conducted at the Northeast Forestry University Experimental Forest Farm, Harbin, Heilongjiang Province, China. The area is located at longitude 126°37′E, latitude 45°43′N, 140 m above sea level, and a slope of 5°. The study area is 43.95 hectares, located in the warm temperate zone semi-humid monsoon climate zone; the annual average temperature is 3.6 °C. The highest temperature in July was 36.4 °C and the lowest temperature in January was −38.1 °C. In the frost-free period, the average rainfall in the area is 600 mm/year. By reforestation between the early 1950s and the end of the 1960s, trees mixed species have been divided into 46 sample plots, each with different type of trees.

In July 2017, in the experimental forest, an experienced forester visually identified 30 Manchurian ash and 30 Populus simonii standing trees that could potentially have internal decay, then 10 solid Manchurian ash and 10 solid Populus trees were also selected, they all were marked in order (Number 1 to number 30 were decay, and number 31 to 40 were solid). All selected trees were between the ages of 50∼60 years old. The DBH (diameter at breast height) of Manchurian ash trees was 20∼38cm, and that of Populus trees was 30∼50 cm.

Nondestructive testing of trees

The instruments used were: PICUS Tree Tronic Electrical Resistance Tomography (ERT) (Argus GmbH, Germany), Arbotom Stress Wave Tomography (SWT) (RINNTECH GmbH, Germany).

ERT measurements were conducted at 100 cm above the ground. The Picus ERT measurement system consisted of 12 electrodes that were evenly placed around the trunk along the horizontal plane during the test. For ease of analysis, the first electrode was arranged in south and the other electrodes were arranged equidistantly in a clockwise manner. Each electrode was clipped and attached to a nail (2 mm in diameter) that had been tightly inserted into the bark and sapwood. After the electrodes were connected, resistance tests were started. In order to reduce the error, two tests were performed at each tree, and the current histograms displayed on the instrument were observed. If there were two differences, a third test must be performed, and then the output file name was recorded for subsequent analysis. Upon completion of ERT measurements, 2D tomograms were obtained combined with the Picus Q72 software program.

After ERT inspections were completed, SWT measurements were carried out (the same positions as for ERT measurement). All sensors were located almost the same as ERT sensors’ positions in the trees, and the transducers were connected at an angle of 90° to the trunk’s longitudinal axis to detect the propagating travel time. Measurements were repeated with the transmitter probe at each point. A complete data matrix was obtained using this measurement process at each test location. These measurements are used as input to the system software. Due to differences in species and paths, two-dimensional (2D) tomographic images were acquired from the original stress wave transmission time using ARBOTOM software to understand the experimental values in this study.

Wood cores obtaining

Wood cores were extracted from the two directions of each cross section using a Swedish wood core drill (diameter 6 mm): one was from the south to the north in the radial direction and the other was from the east to the west in the radial direction, as shown in Fig. 1. When a wood core was decayed, we would get a solid core nearby for comparison. The wood core samples were shown in Fig. 2. After being extracted, the wood cores were immediately put in ziplock bags and were taken back to the laboratory. They were divided into pieces of 1 cm sections, and then they were dried to constant weight at 70 °C in an electric blast oven and weighed.

Figure 1 The direction of wood cores obtaining.

Figure 2 Decayed (A, B) and normal (C) wood core samples.

Calculations and data analysis

Calculation of the true severity of decay

Firstly, mass loss of each wood core was calculated. Dt was defined as the true severity of decay and it was determined by the mass loss of each wood core. Weight per unit length of healthy wood core (ms′) was calculated as (1) ms′=msLs,

where ms′ was the weight per unit length of healthy wood cores (g/cm), ms was the weight of healthy wood cores extracted nearby the decayed wooden core (g), and Ls was the length of the healthy wood cores. If the decayed wood core was still healthy, the estimate weight (md′) was calculated as (2) md′=ms′×L,

where md′ was the weight of wood core if it was still healthy (g), ms′ was the weight per unit length of healthy wood core (g), and L was the length of decayed wood core (cm). The mass loss of each core (Δm) was calculated as (3) Δm=md′−m d,

where Δm was the mass loss of each wood core (g), md′ was the estimated weight of decayed wood core circumstanced healthy (g), md was the actual weight of the decay wood core (g). The severity of decay determined by the mass loss (Dt) was calculated as (4) Dt=Δmmd′×100%

where the Dt was the severity of decay determined by the mass loss of each wood core, the Δm was the mass loss of each core (g), md′ was the estimated weight of decayed wood core circumstanced healthy (g).

Calculation of the severity of decay detected by ERT and SWT

Based upon nondestructive testing of ERT and SWT, the electrical resistance 2D tomograms and stress wave velocity 2D tomograms were acquired. The healthy trees of the same species have the similar 2D tomograms, and decay trees have different 2D tomograms.

In order to quantitatively evaluate the ER tomograms of sample trees, all corresponding electrical resistances (ERs) at each pixel in the tomogram were further calculated by visualization and inversion of the tomograms, and ER maps of the cross-sections were displayed using MatLab software (MathWorks, Natick, MA, USA). The schematic diagram of the ERT and corresponding ER diagram grids are shown in Figs. 3 and 4, respectively. De was defined as the severity of decay detected by ERT, and it was calculated as (5) De=R0−RdR0×100%

where De was the severity of decay determined by ERT, R 0 was the average ER value of the detection direction in the section of the same healthy tree species (Ω), and Rd was the average ER value of the detection direction in the decayed section (Ω).

The velocity distribution of stress waves in a cross-section is shown in Fig. 5. The processing of the stress wave velocity distribution was similar to the ER diagram. Ds was regarded as the severity of decay detected by SWT, and it was calculated as (6) Ds=Vj−VfVj×100%

where Ds was the severity of decay detected by SWT, Vj was the mean velocity of the stress wave in the cross-section of the same healthy tree species (m/s), and Vf was the average velocity of the stress wave in the direction of the decay cross-section (m/s).

Figure 3 Electric Resistance distribution of tree cross-section

Figure 4 The corresponding values of the grayscale (A) and resistance distributions (B).

Figure 5 Velocity distribution of stress waves in a cross-section.

Results

According to the previous calculation and analysis, the severity of decay determined by the wood core mass loss rate was regarded as the true severity of decay (Dt), the severity of decay determined by ERT (De), and the severity of decay determined by SWT (Ds) were calculated, and the results were presented in Table 1. All the data were statistically analyzed, shown in Table 2.

Effectiveness of ERT in detecting decay

Used SPSS (SPSS version 19.0; IBM Corp., Armonk, NY, USA) software to perform the regression analysis of De and Dt, and the analysis results are as follows.

The correlation coefficient (R2) between De and Dt is 0.516 (P < 0.01), and the linear regression equation is (7) De=0.6659Dt+11.852,

when Dt was divided into two parts, Dt < 30% and Dt ≥ 30%, some diffidence was revealed. When Dt < 30%, the correlation coefficient (R) between De and Dt is 0.677 (P <0.01), and the linear regression equation is (8) De=1.3033Dt+4.2855,

when Dt ≥ 30%, the correlation coefficient (R) between De and Dt is 0.300 (P <0.01), and the linear regression equation is (9) De=0.7174Dt+8.2687.

Table 1 Calculation of the test results.

Manchurian ash tree’s Number	Directions	Dt (%)	De (%)	Ds (%)	Populus simonii tree’s Number	Directions	Dt (%)	De (%)	Ds (%)	
1	E-W	20.7	23.6	21.3	1	E-W	65.5	59.8	78.9	
S-N	6.4	15.1	20.2	S-N	60.7	63.6	87.1	
2	E-W	26.1	27.3	30.3	2	E-W	66.5	52	87.3	
S-N	11.1	18	39.2	S-N	62.7	55.2	92.7	
3	E-W	17.1	11.7	32.1	3	E-W	30.5	29.1	41.2	
S-N	10.5	13.3	2.2	S-N	33.8	33.8	57.2	
4	E-W	4.3	2.3	9.2	4	E-W	46.2	30.8	61.3	
S-N	4.7	12.1	15.9	S-N	46.1	38.8	69.1	
5	E-W	3.6	15.8	0	5	E-W	42	36.6	56.2	
S-N	14.6	26.7	5.1	S-N	42.4	25.5	63.9	
6	E-W	7.6	8	14.9	6	E-W	37	21	46.5	
S-N	7.6	6.4	12.9	S-N	36.7	45.9	29.7	
7	E-W	13.5	24	0	7	E-W	38.6	32.9	34.3	
S-N	5.9	14.1	18.8	S-N	46.5	38.8	55.4	
8	E-W	3.7	6.5	0	8	E-W	34.7	33.5	32.2	
S-N	14.7	20.1	10.2	S-N	41.9	50.7	44.1	
9	E-W	0.4	3.7	0	9	E-W	44.2	31.8	63.2	
S-N	7.2	14.7	5.7	S-N	40.3	47	41.7	
10	E-W	28.3	32.9	42.3	10	E-W	43.6	58	54.2	
S-N	16.7	45.5	57.1	S-N	31.7	25.3	46.6	
11	E-W	14.5	21.1	46.3	11	E-W	35	40.3	48.6	
S-N	5.8	17.4	25.3	S-N	50	46.4	46.9	
12	E-W	28.9	55.8	79.2	12	E-W	33.9	34.3	35.5	
S-N	3.1	2.5	1.2	S-N	36.4	43	30.3	
13	E-W	1.3	4.6	1.1	13	E-W	58.6	25.6	57.6	
S-N	3.5	3.8	4	S-N	56.9	57	65.2	
14	E-W	7.3	15	14.3	14	E-W	38.6	42.7	47	
S-N	12.2	20.5	19.2	S-N	39.7	35	45.5	
15	E-W	10	21	43.1	15	E-W	38.5	28	36.5	
S-N	15.2	16.5	10.2	S-N	39.4	49	35.2	
16	E-W	23.5	43.6	40.1	16	E-W	35.9	24	40	
S-N	17.6	27	26.8	S-N	40	21	42	
17	E-W	5.4	19.9	24.2	17	E-W	35.9	35.8	37.5	
S-N	11.2	13.2	11.2	S-N	43.3	44	32.9	
18	E-W	17.3	15.3	21.3	18	E-W	43.3	50.6	33.8	
S-N	27.4	30.3	60.8	S-N	42.3	33.9	43.5	
19	E-W	28.4	53.2	55.8	19	E-W	39.6	21	64.7	
S-N	6.3	29.8	35.2	S-N	31.9	30	32.9	
20	E-W	15.9	40.4	44.4	20	E-W	30.1	32	25.7	
S-N	25.8	45.6	52.3	S-N	36.1	14.2	26.1	
21	E-W	21.7	47.4	50.6	21	E-W	32.6	14.9	39.3	
S-N	16.3	20	27.8	S-N	50.2	34.5	59.3	
22	E-W	24.7	29	35.8	22	E-W	47.5	43.5	53	
S-N	26.9	25.1	11.3	S-N	43.7	45.6	45.3	
23	E-W	26	25	28.9	23	E-W	42	42	40.6	
S-N	24.5	35	30.3	S-N	39.8	36.1	41.3	
24	E-W	29.7	40	31.3	24	E-W	32.6	31	39.1	
S-N	28.6	43.3	33.2	S-N	55	26.3	55.9	
25	E-W	29	41	38.4	25	E-W	34.6	49	44.5	
S-N	27.3	51	28.3	S-N	35.8	39.3	35.8	
26	E-W	29.4	52.4	21	26	E-W	37.5	30	37.5	
S-N	30.7	39.9	35.2	S-N	40.8	41.2	40.8	
27	E-W	37	31.6	46.1	27	E-W	36	29.3	35.1	
S-N	54.8	52.3	71.6	S-N	33.5	57.5	32.2	
28	E-W	40.5	21.2	48.1	28	E-W	35.9	47.3	32.4	
S-N	35.5	36.6	40.3	S-N	38	27	37.3	
29	E-W	31.6	35	49.3	29	E-W	42	34.4	39.4	
S-N	34.2	40.3	56.4	S-N	35.4	25	34.1	
30	E-W	59.9	59.3	67.5	30	E-W	33.1	22	33.2	
S-N	64.9	66	75.6	S-N	38.5	36.2	35.5	
Notes.

Dt was the severity of decay determined by the mass loss; De was the severity of decay determined by ERT; Ds was the severity of decay determined by SWT.

table 2 Statistical analysis of various test results.

cay degree (%)	Category	Average value	Maximum value	Minimum value	Standard deviation	Skewness	Kurtosis	Normality test	
Dt < 30	De	22.82	55.80	2.30	14.83	0.49	−0.70	followed normal distribution	
Ds	24.46	63.90	0.00	17.26	0.79	−0.13	followed normal distribution	
Dt ≥ 30%	De	35.51	58.00	14.20	9.78	0.06	−0.21	followed normal distribution	
Ds	45.50	76.00	19.50	10.04	0.51	0.16	followed normal distribution	
Notes.

Dt was the severity of decay determined by the mass loss; De was the severity of decay determined by ERT; Ds was the severity of decay determined by SWT.

These results are plotted in Fig. 6.

Figure 6 The scatter plot between result of resistance De and the true severity of decay Dt.

Effectiveness of SWT in detecting decay

SPSS software was also used to conduct the regression analysis of Ds and Dt, and the analysis results are the following.

The correlation coefficient (R) between Ds and Dt is 0.638 (P < 0.01), and the linear regression equation is (10) Ds=0.9993Dt+7.5369,

when Dt < 30%, the correlation coefficient (R) between Ds and Dt is 0.398 (P< 0.01), and the linear regression equation is (11) Ds=1.2501Dt+5.9497,

when Dt ≥ 30%, the correlation coefficient (R) between Ds and Dt is 0.645 (P <0.01), and the linear regression equation is (12) Ds=1.3441Dt−8.0242,

These results are plotted in Fig. 7

Figure 7 The scatter plot between result of stress wave Ds and true severity of decay Dt.

Discussion

Analysis of the relationship between De and Dt

Once the wood is infected by wood-destroying rotten, its cell walls are decomposed and cause the wood to rot and disintegrate. When the wood is rotted and discolored, the hypha growth requires a lot of water, which will increase the moisture content of the decayed area, and then ions will be released from the wood cells. Studies (Houston, 1971) have shown that with the discoloration and decay of standing trees, the content of metal ions such as potassium, calcium, manganese, and magnesium in the rotten wood increase. As the concentration of cations increased, the electrical resistance of decayed and discolored wood was significantly reduced compared to healthy wood (Ostrofsky, Jellison & Shortle, 1997; Nilsson, Karltun & Rothpfeffer, 2002; Jonàs, Carmen & Jan, 2011). The severity of decay detected by ERT (De) mainly reflects the increase of the moisture content and metal ions in the decayed trees (Bieker & Rust, 2010). The severity of decay determined by mass loss rate of wood (Dt) mainly reflects the wood weight loss rate, which is closely related to decay distribution range, wood structural damage, and mechanical strength. Both De and Dt reflect the different stages of decay in wood. In this study, when Dt < 30%, there was a significant correlation between De and Dt, and the correlation coefficient was the highest. Therefore, ERT can make a good diagnosis in the early stages of decay in trees. If ERT is used to detect wood in the early stages of decay, so we can know the condition of the trees early and can deal with the damage caused by decay as soon as possible.

Analysis of the relationship between Ds and Dt

If a large amount of cellulose, hemicellulose, and lignin in wood are corroded by woody rot, decay will occur, and the density will decrease accordingly, and defects will form inside the wood. When the stress wave propagates in the defective wood, it propagates along the edge around the defect. The propagation path changes from a straight line to a curved line. The propagation time increases and the speed decreases (Xu, Xu & Wang, 2014). The severity of decay detected by stress wave (Ds) mainly reflects the size of the internal defects of the standing tree (Tannert et al., 2014), while the mass loss rate of wood is also closely related to the range of decay, the stage of damage to the structure of the wood, and the mechanical strength. Therefore, both methods can reflect the decay status of standing trees, so there is a correlation between them. In this study, when Dt ≥ 30%, there is a significant correlation between Ds and Dt, and the correlation coefficient is higher than the correlation between De and Dt. In other words, in terms of decay degree, SWT is a better indicator than ERT when Dt ≥ 30%.

Comparison of two NDT methods

Electrical resistance value of standing trees is affected by many factors such as environmental humidity, temperature, moisture content, the decay stage, growth season, and measurement site, etc., which will affect the measurement effect of the resistance value, moreover, it is easy to misjudge for the sensitivity of resistance testing (Just & Jacbbs, 1998; Wang, Yang & Xu, 2001). Stress wave detection results are influenced by factors such as cross-sectional shape, decayed severity and number of sensors (Gilbert & Smiley, 2004).

In different stages of decay, for these two methods, when Dt < 30%, De had a relatively high correlation with Dt. When Dt ≥ 30%, Ds had a relatively high correlation with Dt. These results are related to the decay process of timber. During the early stage of decay, wood quality and visual appearance look unchanged; however, the chemical components have changed markedly. Wood decay fungi can be propagated through the extension and spread of mycelium or mycorrhizae fungi. When wood decay fungi enter wood cells and settle between wood cells, they secrete various enzymes to decompose cellulose, hemicellulose, and lignin in the cell walls of the wood into sugars, which are further digested as nutrients (Van der Wal, Ottosson & De Boer, 2015). Electrical resistance of wood is mainly related to moisture and metal-ion content, so ERT was more accurate during the early stage of decay. Once decomposition of fungi began to stabilize, the variation of electrical resistance became to flatten. During early stage of decay, SWT was inaccurate owing to the lack of cavities (Xu, Xu & Wang, 2014). When cavities were present, SWT was more accurate, since, with the slow growth of holes, the decay grew worse (Li, 2014). Some research has shown that a stress wave could travel around the holes when it encountered them, and then the propagation of the stress wave would lengthen and its travel time increase. According to the calculation method of stress waves, the velocity change of a stress wave could reflect the severity of decay (Tannert et al., 2014).

Conclusions

The purpose of this study was to compare two non-destructive testing methods to determine the technology that matches the specific conditions in the forest. Eighty live trees were tested using three methods: Electric resistance tomography, stress wave tomography, and mass loss ratios of wood increment cores. The results were shown as follows:

1. There was a clear positive correlation between the severity of decay detected by ERT (De) and the true severity of the decay (Dt). When Dt < 30%, De had the higher correlation coefficient (R2 = 0.677, P <0.01) with Dt than SWT method.

2. (An obvious positive correlation was shown between the severity of decay detected by SWT (Ds) and the true severity of decay (Dt), and when Dt ≥ 30%, Ds had the higher correlation coefficient (R2 = 0.645, P <0.01) with Dt than ERT method.

3. Both ERT and SWT could characterize the wood mass loss rate, which is the index for expressing the severity of decay. Therefore, two NDT methods can effectively detect the decay of standing trees in certain stages of decay.

4. ERT and SWT had distinctive features and advantages. ERT can give a better diagnosis than SWT for the early stage of decay (Dt < 30%) in standing trees, while SWT can be effectively used in the late stage of decay (Dt ≥ 30%). Therefore, it is suggested that each technique can be employed in practical internal decay testing for standing trees according to decay stage and operational conditions.

In addition, it should be mentioned that the purpose of this paper is to compare the two techniques to take advantage of individual decay detection capabilities. However, it should be noted that internal imaging can provide valuable information about the location and extent of decay that simple core drilling cannot provide.

Additional Information and Declarations

Competing Interests

Author Contributions

Field Study Permissions

Data Availability

The authors declare there are no competing interests.

Xiaoquan Yue conceived and designed the experiments, performed the experiments, analyzed the data, authored or reviewed drafts of the paper.

Lihai Wang conceived and designed the experiments, contributed reagents/materials/analysis tools, authored or reviewed drafts of the paper, approved the final draft.

James P. Wacker approved the final draft.

Zhiming Zhu performed the experiments, analyzed the data, prepared figures and/or tables.

The following information was supplied relating to field study approvals (i.e., approving body and any reference numbers):

Northeast Forestry University

The following information was supplied regarding data availability:

The raw measurements are available in Tables 1 and 2.

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
