# Peer review of "Electric resistance tomography and stress wave tomography for decay detection in trees—a comparison study"

_PeerJ, doi:10.7717/peerj.6444_

## Round 0.1 · original submission · Minor Revisions

The reviews validate the technological value of the manuscript. However, significant improvements have been suggested by the reviewers mainly related to English language usage and outdated citations. Each reviewer has identified specific revisions either in comments or as annotated within the reviewing manuscript. You have here an opportunity to upgrade the quality of this manuscript for publication in PeerJ. Please carefully address each point raised by the reviewers in a separate rebuttal letter and provide tracked revisions in the word doc. Also, please address the rather negative perspective of reviewer #1. We look forward to receiving your revised manuscript.

Reviewer 1 ·

Basic reporting

The writing of this paper is neither clear nor unambiguous. It falls so far short of being understandable that this reviewer feels unable to even begin to review the validity of the science.
The references are not properly cited in the text, nor accurately listed in the bibliography.

The authors would be advised to seek professional help in rendering this paper into "clear and unambiguous, professional English." Alternatively, perhaps the paper's third author, based in the United States, could take the lead on this.

Experimental design

It is unclear why the authors, in choosing to use the Argus TreeTronic equipment for Electrical Resistance Tomography, did not use the company's PiCUS equipment for sonic tomography, their version of the Rinn Stress Wave Tomography. This does not appear to be discussed in the paper, nor referenced, even though there are numerous publications demonstrating its use.

Validity of the findings

Given the inscrutable writing of this paper, this reviewer feels unable to properly evaluate the paper. That said, the experiment approach, while interesting, appears flawed, in using wood cores, which provide data in a single dimension.

Reviewer 2 ·

Basic reporting

A significant proportion of the literature cited is a bit outdated. Citations to more recent articles should be increased.
Line 246 must be corrected: (Van der Wal et al, 2015)
Figures 1 and 2 do not contribute anything to the understanding of the article.
Figures 3, 5, 6 and 7 should be summarized in a single figure that synthesizes the information provided.
Appendix 1 should not be an integral part of the article and be part of a complementary file.

Experimental design

The methods are described in detail and work could be replicated.
The sample size of the experiment is adequate.

Validity of the findings

It is an article that does not present a high of novelty neither in its methodology nor in its obtained results. Nevertheless, the main finding is the detection of a threshold of 30% in the true severity of the tree decay that allows one or another technique to become more effective.
This conclusion is robust according to the results.

Additional comments

I recommend this article for publication, subject to an increase in the relevance of the figures and an improvement in the proportion of relevant citations of the last 5 years.

Reviewer 3 ·

Basic reporting

This article is clear and unambiguous with well understand of English usage.
The references used were from a multiple sources.
The article structure, figures and table are well positioned significantly.

Experimental design

The experimental design had been well explained with sufficient information.

Validity of the findings

The findings presenting in this article show the scientific merit and contribution to the knowledge.

Additional comments

The manuscript had been well written with easy understanding and clear explanation for the reader. Anyhow, in my point of view there is a comment/suggestion in the article which can be improved to enhance the quality of this paper.

Annotated reviews are not available for download in order to protect the identity of reviewers who chose to remain anonymous.

·

Basic reporting

The structure, aims, description and result reporting are good quality, suitable for publication. Literature review is thorough and adequate.

English usage is, albeit not flawless, mostly adequate, easy to follow and understand. There are some exceptions to this that are marked in the review material. These problems should be corrected.

Figures are relevant and mostly easy to understand. All three trendlines should be included in figures 8 and 9 to make the analysis easier to follow for the reader.

Results are self-contained and relevant, although the analysis could have a somewhat broader outlook to the application of NDT methods (see note at conclusions.)

Experimental design

Paper contains significant and relevant primary research in the area of biological research. The aim of the research is well-defined, relevant, and fills an identified knowledge gap. Research is carried out rigorously and yielded trustworthy results. Methods are described in sufficient detail; the description of the calculation should be improved as noted.

Validity of the findings

Result analysis is thorough, statistically sound and relevant. Conclusions are well stated, but a note regarding the wider applications of the discussed tomographic methods should be included, see note in the review material.

Additional comments

Minor notation mistakes are also indicated in the text.

---

## Round 0.2 · accepted · Accept

Thank you for the revised manuscript. Your manuscript is now accepted for publication in PeerJ.

Reviewer 2 ·

Basic reporting

The manuscript is clear, Literature references are adequate.

Experimental design

The methods are described in detail and work could be replicated. The sample size of the experiment is adequate.

Validity of the findings

The results are a contribution to a field where new scientific communications do not usually bring novelties.

Additional comments

Homogenize the precision in table 1